# Onchocerciasis-associated epilepsy in Maridi, South Sudan: Modelling and exploring the impact of control measures against river blindness

Samit Bhattacharyya [1,2]*, Natalie V. S. Vinkeles Melchers[3], Joseph N. Siewe Fodjo[2], Amit Vutha[4], Luc E. Coffeng[3], Makoy Y. Logora[5], Robert Colebunders[2], Wilma A. Stolk[3]

1 Department of Mathematics, School of Natural Sciences, Shiv Nadar Institution of Eminence, Dadri, Uttar Pradesh, India, 2 Global Health Institute, University of Antwerp, Antwerp, Belgium, 3 Department of Public Health, Erasmus MC, University Medical Center Rotterdam, Rotterdam, The Netherlands, 4 Department of Mathematics, Ohio State University, Columbus, Ohio, United States of America, 5 National Neglected Tropical Disease Programme, Ministry of Health South Sudan, Juba, South Sudan

* samit.b@snu.edu.in

**Data Availability Statement:** The simulation codes are available in the supplementary appendix, and also through GitHub: https://gitlab.com/

## Abstract

### Background

Onchocerciasis, also known as "river blindness", is caused by the bite of infected female blackflies (genus *Simuliidae*) that transmit the parasite *Onchocerca volvulus*. A high onchocerciasis microfarial load increases the risk to develop epilepsy in children between the ages of 3 and 18 years. In resource-limited settings in Africa where onchocerciasis has been poorly controlled, high numbers of onchocerciasis-associated epilepsy (OAE) are reported. We use mathematical modeling to predict the impact of onchocerciasis control strategies on the incidence and prevalence of OAE.

### Methodology

We developed an OAE model within the well-established mathematical modelling framework ONCHOSIM. Using Latin-Hypercube Sampling (LHS), and grid search technique, we quantified transmission and disease parameters using OAE data from Maridi County, an onchocerciasis endemic area, in southern Republic of South Sudan. Using ONCHOSIM, we predicted the impact of ivermectin mass drug administration (MDA) and vector control on the epidemiology of OAE in Maridi.

### Principal findings

The model estimated an OAE prevalence of 4.1% in Maridi County, close to the 3.7% OAE prevalence reported in field studies. The OAE incidence is expected to rapidly decrease by >50% within the first five years of implementing annual MDA with good coverage (≥70%). With vector control at a high efficacy level (around 80% reduction of blackfly biting rates) as the sole strategy, the reduction is slower, requiring about 10 years to halve the OAE

erasmusmc-public-health/wormsim2.78src2. After de-identification all individual participant data underlying the results reported in this article is available via the Zenodo repository at https://zenodo.org/record/7932357#.ZF9JBi1h1TY.

**Funding:** RC gratefully acknowledge the European Research Council (ERC) Advanced grant (NSETHIO project No. 671055). SB received funding from Science and Engineering Research Board (SERB) India (Grant No. SB/S9/Z-11/2017) for travel support and visiting Global Health Institute, University of Antwerp. The funders had no role in study design, data collection and analysis, decision to publish, or preparation of the manuscript.

**Competing interests:** The authors have declared that no competing interests exist.

incidence. Increasing the efficacy levels of vector control, and implementing vector control simultaneously with MDA, yielded better results in preventing new cases of OAE.

## Conclusions/Significances

Our modeling study demonstrates that intensifying onchocerciasis eradication efforts could substantially reduce OAE incidence and prevalence in endemic foci. Our model may be useful for optimizing OAE control strategies.

## Author summary

Onchocerciasis is a parasitic disease caused by the filarial worm *Onchocerca volvulus*, which is transmitted by *Simuliidae* black flies. This disease is most common in Africa and South America. *Onchocerca volvulus* is thought to infect approximately 35 million Africans these days. Adult female worms infect people and form subcutaneous nodules, releasing thousands of microfilariae per day, causing itching, dermatitis, blindness, and epilepsy. Despite the fact that the association between onchocerciasis and epilepsy was already mentioned in a study from Mexico in 1938, this association was only reported around 1991 in Africa in the Mbam valley in Cameroon. Onchocerciasis-associated epilepsy (OAE) is characterised by seizures that start between the ages of 3 and 18 years in previously healthy children with a high degree of *O. volvulus* infection. Raising awareness about OAE will increase adherence to community-directed treatment with ivermectin (CDTI), particularly in high-prevalence areas, while also motivating funders and stakeholders to keep fighting onchocerciasis. Here, we create an OAE model within the framework of ONCHOSIM that is parametrized using South Sudan data and investigate how different control measures such as CDTI and vector control might help reduce OAE in the future.

## Introduction

Epilepsy is a chronic neurological disorder with important psychosocial and economic consequences. Around 50 million people worldwide are estimated to have epilepsy and nearly 80% of them live in low- and middle-income countries, especially in Sub-Saharan Africa (SSA) [1]. The median prevalence of epilepsy in SSA is 1.4%, almost doubling the global epilepsy prevalence estimated at 0.76% [2,3]. Risk factors for developing epilepsy include complications during birth, traumatic brain injury, cerebral vascular disease, brain tumors, and infections of the central nervous system (CNS) [3]. In SSA, parasitic infections, such as cerebral malaria and neurocysticercosis, are additional causes of epilepsy [4,5].

There is increasing evidence suggesting that onchocerciasis, besides its well-known skin and ocular manifestations [6,7], may also cause epilepsy by a yet to be identified mechanism [8]. Onchocerciasis, also known as river blindness, is a parasitic disease caused by the filarial worm *Onchocerca volvulus* transmitted by blackflies of the genus *Simuliidae* [6,9]. Although the infection exists in some regions of Latin America and Yemen, more than 99% of infected people live in Africa [10]. A high prevalence of epilepsy has been observed in several onchocerciasis-endemic regions of SSA such as in South Sudan [11,12], Cameroon [13], and Tanzania [14]. Nodding syndrome, a particularly devastating form of epilepsy characterized by nodding seizures and stunting, is likely also associated with onchocerciasis [15].

Many epidemiological studies have documented an association between onchocerciasis and epilepsy. Pion et al. [16] assessed the relationship between the prevalence of onchocerciasis and epilepsy using available population-level data from different countries in SSA. They calculated that epilepsy prevalence increases by 0.4% for each 10% increase in onchocerciasis prevalence. The studies by Chesnais et al [17,18] showed that the risk to develop epilepsy increases with higher *O. volvulus* microfilariae (mf) intensities. They thus found a dose-response relationship between mf density and epilepsy risk in onchocerciasis-infected children, suggesting that epilepsy in this population is directly or indirectly caused by *O. volvulus* infection.

Large-scale interventions programs are ongoing in endemic countries to control and eliminate onchocerciasis. Ivermectin mass drug administration (MDA) is the main approach to quell onchocerciasis in endemic foci [19,20], sometimes in combination with localized vector control. Several studies have shown that interventions against onchocerciasis can lead to a reduction in the incidence of onchocerciasis-associated epilepsy (OAE) [21,22]. Community-based studies conducted in 2017 in a highly affected area in Northern Uganda found that, compared to baseline findings in 2012, the incidence of nodding syndrome (OAE with head nodding seizures) and other forms of epilepsy drastically declined from 1,165 to 130 per 100,000 persons (P = 0.002) in an area with intensive interventions (annual MDA since 2009, biannual MDA since 2013, ground-based larviciding since 2012). Similarly, no new cases of OAE have appeared in the Kabarole district of Western Uganda after onchocerciasis was eliminated from the area [23].

Despite the overall success of onchocerciasis control and elimination programs, there are still many onchocerciasis-endemic regions with high ongoing *O. volvulus* transmission and most likely high OAE disease burden. It was estimated that by the year 2015, about 381,000 persons (95% CI: 159,000–1,636,000) were suffering from OAE across onchocerciasis-endemic areas in central and eastern Africa [24]. In the Maridi County in South Sudan (located on the border with DRC), where skin snip studies among randomly selected adults without epilepsy revealed an onchocerciasis prevalence of 50% [15], a recent survey found an overall epilepsy prevalence of 4.4% and an annual incidence of 374 per 100,000 persons [12]. Individuals between 11 to 20 years old were most affected (age-specific prevalence of 10.5%). The epilepsy prevalence was also highest in the villages close to the Maridi dam, a blackfly-breeding site. This dam, constructed in 1954–55 and repaired in 2000, may explain why an increasing number of children in Maridi have been developing OAE since the early 2000s [25].

Mathematical modelling has been used, among others, to assess prospects for eliminating onchocerciasis by annual ivermectin MDA or alternative strategies [26–28] and has also been used to study likely trends in morbidity following the introduction of MDA [29,30]. So far, modelling has not yet been used to predict the impact of onchocerciasis elimination efforts on the incidence and prevalence of OAE. In the current paper, we used the modelling framework ONCHOSIM (a mathematical model for simulation of onchocerciasis transmission and control [31,32]) to investigate the impact of control interventions on the incidence and prevalence of OAE.

## Methods

### Ethics statement

Ethical approval was obtained from the ethics committee of the Ministry of Health of South Sudan (January 2018, MOH/ERB 3/2018) and the University of Antwerp, Belgium (April 2019, B300201940004). Signed or finger-printed informed consent was obtained from all participants, parents or Care Givers, and assent was also obtained from adolescents (aged 12–18 years). All personal information was encoded and treated confidentially.

## Onchocerciasis and epilepsy data used for modelling

We used age-specific OAE prevalence data collected from villages in the Maridi County (Western Equatoria State, South Sudan) [12,37] for parameter estimation of the model. This area is known to be endemic for onchocerciasis and the pre-control nodule prevalence was estimated at about 30–40% in the late 1990s [42], corresponding to a meso-endemic status. However in view of the high mf and OV16 prevalence as well as high biting rates observed at Maridi in 2018 and 2019 [15,25] (details below), we surmise that there has been an increase in onchocerciasis transmission since the 1990s. Consequently, in this modelling paper we consider this focus as being currently hyper-endemic.

MDA was initiated in Maridi County in 1997, but it was not conducted every year due to intermittent violent conflicts [43], with only six rounds having taken place between 1997 and 2013 [44]. Regular ivermectin MDA was only resumed in 2017, when 40.8% of the population reported to have taken the drug [12]. In May 2018, a door-to-door survey was carried out in this area to identify people with epilepsy. A total of 2,511 households (n = 17,652 participants) were surveyed and 774 persons (4.4%) persons with epilepsy were identified [12]. About 85% of the examined persons with epilepsy fulfilled the OAE criteria [37], suggesting an OAE prevalence of 3.7%. Only persons with epilepsy meeting the OAE criteria [38] were counted as "cases" in the model presented in this paper. In December 2018, the mf prevalence in Maridi was 50% among 50 non-epileptic adults tested and 85% among 270 persons with epilepsy tested [15]. In November 2019, biting rates up to 202 flies/man/hour were noted close to the Maridi dam and an onchocerciasis seroprevalence of 66.7% was documented using the OV16 rapid test among children 7–9 years old [25]. These figures suggest that currently, there is high ongoing onchocerciasis transmission in the Maridi area. It is very unlikely that neurocysticercosis could be an etiology for epilepsy in these villages, as there are no pigs in Maridi (for cultural reasons) [12].

## ONCHOSIM model

ONCHOSIM is a well-established stochastic mathematical model that is implemented in WORMSIM, a generic individual-based helminth modelling framework [26,32,45]. We used version 2.78 of the WORMSIM program, which incorporates an elaborate morbidity sub-model that allows for the simulation of a wide spectrum of clinical manifestations due to *O. volvulus* infection [29,30]. A mathematical description of this module and the specific details of the disease parameters and processes of disease development can be found elsewhere [30].

## Model assumptions

We assume that brain damage is (directly or indirectly) induced by immune responses triggered by *O. volvulus* mf, and that the accumulated amount of brain tissue damage in individual $i$ at time $t$, $D_i(t)$, is given by:

$$D_i(t) = D_i(t-1) + S_i d_i(t)$$

where $S_i$ is the susceptibility index of an individual $i$ for developing brain tissue damage, and $d_i$ is the number of mf that die in individual $i$ in month $t$. The damage is assumed to be irreversible. When $D_i(t)$ exceeds a threshold, a person is assumed to acquire OAE, which is irreversible. As the age of onset of epilepsy between three and 18 years is characteristic of OAE (Box 1), we introduced a new parameter in the disease sub-model, $a_{max}$, to define a maximum age for mf accumulation beyond which onset of OAE is no longer possible. However, the age of onset of OAE is not exactly known. For this paper, we fixed it at 18 years, varying it in a

Box 1: *OAE age patterns*

Population-based epilepsy surveys in onchocerciasis-endemic regions, including in the Maridi area, show that a large number of individuals develop epilepsy between the ages of three and 18 years, with a peak age of onset around 10 years [33–37] (Fig 1A). This age range of onset of epilepsy is one of the main criteria of the definition of OAE [38]. In non-onchocerciasis endemic areas in Africa, epilepsy incidence usually peaks before the age of 5 years due to perinatal causes or epilepsy of genetic origin [4,8] (Fig 1B). Nodding syndrome is a more severe form of OAE with an earlier onset of seizures around the age of eight years, while for other forms of OAE, this is generally around the age of 11–12 [15,34]. Persons with nodding syndrome also have higher mf densities than those with other forms of OAE [15]. Nodding syndrome onset after the age of 20 years has so far not yet been reported. The reason for this peak onset around 10 years may be the sharp increase of mf load in *O. volvulus* infected children prior to this age [39] and also an increased susceptibility of the brain around this age to develop epilepsy, as in juvenile epilepsy [40]. The actual pathophysiological mechanism of OAE is still under investigation [8,41].

sensitivity analysis. Each individual was assigned a random, susceptibility index $S_i$ in $[0, \infty)$, drawn from a gamma distribution with mean one (i.e., shape = rate in gamma distribution). Depending on their susceptibility index, some individuals may develop epilepsy very early in life compared to others who may not at all develop epilepsy in their lifetime at a given *O. volvulus* mf load.

It has also been observed that there is a relatively high mortality rate of persons with epilepsy in onchocerciasis-endemic areas compared to the general population [12,37,46]. In fact, most people with epilepsy in South Sudan die before the age of 20 years and none of the identified cases were older than 35 years [12]. This is captured by assuming excess mortality in OEA patients, modelled as a reduction in a person's residual life expectancy after developing OAE. The relative reduction per individual is randomly sampled from a uniform distribution.

## Parameter quantification

We fixed most demographic and biological parameters related to fly and worm at their default value, as used in previous work, but we simulated a much larger population size to obtain more reliable output on the incidence of OAE by age (see Section A in S1 Text, and supplementary text S1 of [30]). Next, we quantified parameters in ONCHOSIM related to local transmission conditions and individuals' exposure to blackflies, which largely determine the modelled equilibrium endemicity level. We fixed parameters related to age-sex and random variation in exposure to fly bites to the default values that were used previously [30]: exposure to blackfly bites is zero at birth, increases linearly with age until age 20, and remains stable at this maximum level at higher ages. The mean number of bites in women was assumed to be 70% of the number of bites in men in all ages. Random, age-sex independent variation in exposure to fly bites between individuals is described by a gamma distribution with mean 1.0 and variation 1/k, with *k* (shape and rate) equal to 4.283 similar to previous simulation exercises with ONCHOSIM for hyperendemic areas. Only the Annual Biting Rate (ABR) was tuned to have the model mimic the pre-control mf prevalence that was observed in Maridi, by adjusting the relative biting rate (*rbr)* that is multiplied with a series of monthly biting rates (for January to December) to obtain the ABR in the simulated population. As previously mentioned, we

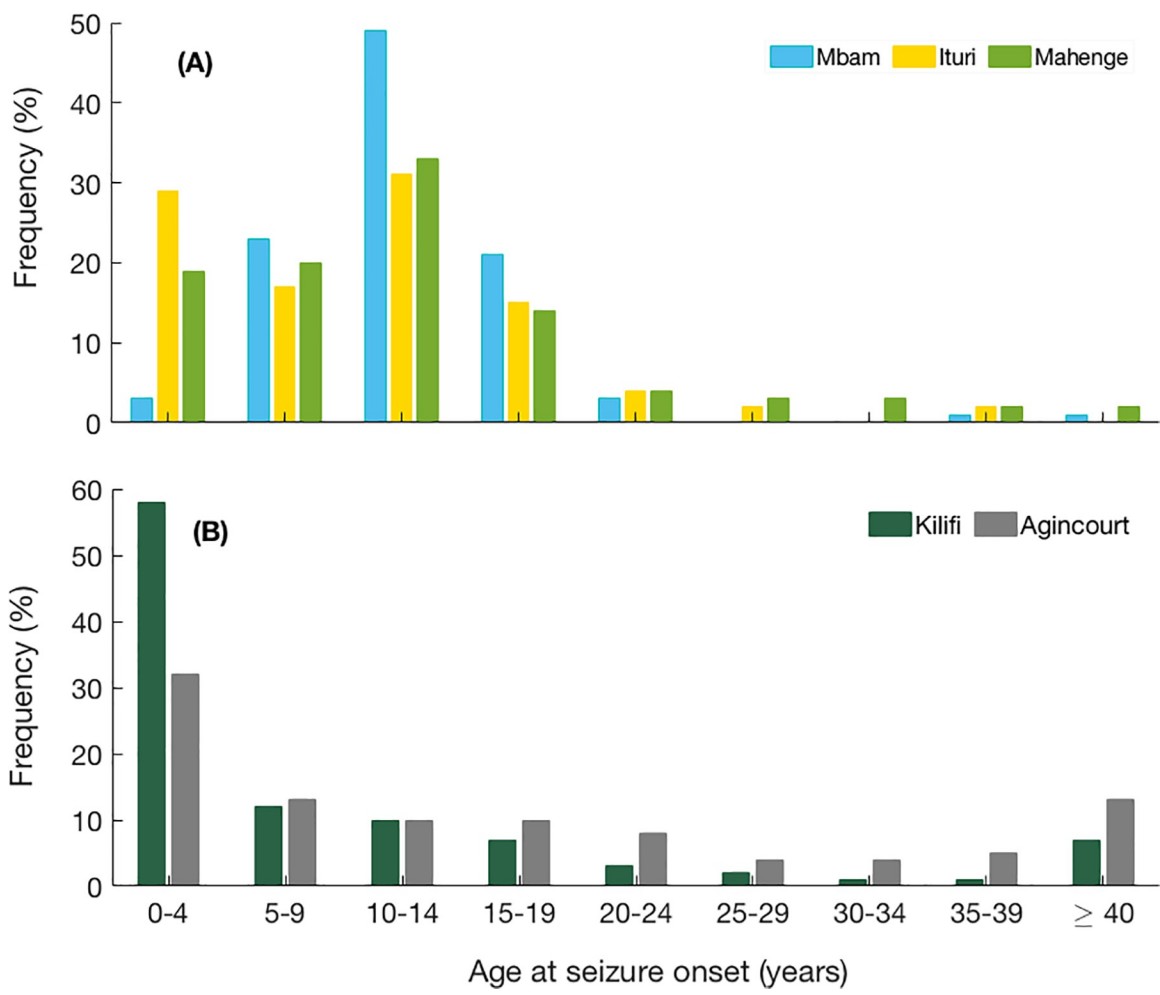

**Fig 1.** Age of onset of epilepsy in A) onchocerciasis-endemic villages: Mbam (Cameroon), Ituri (Democratic Republic of Congo), Mahenge (Tanzania), and B) onchocerciasis non-endemic villages: Kilifi (Kenia), Agincourt (South Africa) as previously reported by [8].

consider Maridi as a hyperendemic focus considering our recent study findings in the Maridi villages [15,25]. Accordingly, we calibrated the *rbr* in the model (that represents a pre-control equilibrium situation) such that with a larger number of model-runs, our model could adequately reproduce a mean mf prevalence that fits within the range that describes onchocerciasis hyperendemicity as observed in the Maridi area.

We fitted model-generated age-specific OAE prevalence to the age-specific OAE prevalence data from Maridi, to estimate three parameters of the model, i.e. (i) the disease threshold ($T$), (ii) shape parameter of the gamma distribution describing inter-individual variation in susceptibility for OAE ($S_i$), and (iii) the mean life expectancy reduction (LER) experienced by people developing OAE. In estimating the parameter LER, we estimated Minimum LER, and calculated the maximum LER by considering Maximum LER = Minimum LER+10, and then determined the mean.

We used orthogonal sampling (Latin Hypercube sampling [LHS]) technique in 3D-parameter space and minimized the least square statistic obtained from data and the model output. Starting from wide range of each of the parameters, we sampled 100 points in three-dimensional parameter space using LHS, and then calculated the sum of squared errors (SSE) of age-

specific OAE prevalence obtained from the model and South Sudan data. We considered the parameter combination which produced smallest SSE out of all these 100 samples. Thereafter, we took small neighborhoods around these parameter values and repeated the LHS to estimate the SSE. To reduce the impact of stochastic variability, we performed 100 runs for each set of parameter values and fitted the average model-predicted age-specific OAE prevalence to the data. We continued this execution of sampling and running the simulation iteratively multiple times until we observed no significant changes in the SSE. We could then obtain the parameter combination as their final estimated values explaining the South Sudan data. A supplementary figure has been given how SSE values depends on the values of three parameters (Fig A in S1 Text).

To validate the fitting exercise, we plot model-predicted OAE prevalence as a function of total mf prevalence. We simulated the model (with estimated parameters) for different values of monthly biting rate by varying the ABR from 6,170 to 30,850, and then calculate the OAE prevalence to plot against the MF prevalence (see results).

## Sensitivity analysis

We performed univariate sensitivity analysis to test the sensitivity of parameter estimation exercise to assumptions on: (i) $a_{max}$, the parameter describing maximum age of mf accumulation to develop epilepsy; (ii) inter-individual variation in susceptibility to develop OAE, (iii) reduction in life expectancy of epileptic individuals, and (iv) ABR. We used the same LHS technique to draw 100 samples from a wide range of the respective parameter around the best fitted values. For example, $a_{max}$ is sampled from 15–20, the shape parameter sampled from 0.075 to 0.2, the reduction in mean residual life expectancy parameter varies from 60%-80% [35], and the ABR from [20,000; 40,000].

## Predicting trends in morbidity with onchocerciasis control

Scenario analyses were performed to evaluate the impact of annual and biannual ivermectin MDA on OAE incidence and prevalence over time, varying the MDA duration from 10–20 years and coverage between 50% and 70%. Of note, the denominator used for the calculation of MDA coverage was the entire population (all ages included). In addition to MDA, we also ran simulations for the implementation of vector control for 20 years with varying effectiveness (percentage reduction of blackfly biting rates) as standalone strategy or in combination with MDA. For each scenario, we computed the mean predicted OAE incidence out of 100 repeated stochastic simulation runs to avoid stochastic variability. We further perform multivariate analysis to assess the sensitivity of model-predicted trends in OAE incidence to changes in values for the OAE model parameters. We used same Latin Hypercube sampling [LHS] technique in 3D-parameter space considering feasible range of the estimated values of the three parameters: (i) inter-individual variation in susceptibility to develop OAE [0.075–0.2], (ii) reduction in life expectancy of epileptic individuals [60%-80%], and (iii) the Mf threshold [600–800]. We draw 100 samples from the above range of the respective parameters, and run the model under different MDA coverage, and vector control.

## Good modelling practice

Our adherence to the five principles of the NTD Modelling Consortium on good practice for policy-relevant modelling is described in S1 Table.

## Results

### Parameter estimation

The pre-control model of OAE fits well to the age-specific pattern of the South Sudan OAE prevalence as observed in Fig 2. The *rbr* was estimated at 0.91, resulting in an ABR of 0.91 * 30,850 = 28,074 and a mean mf-prevalence of around 78.4% in the entire population. The morbidity parameter values were estimated as follows: mf threshold count ($T$) = 721, susceptibility heterogeneity parameter ($S_i$) = 0.137, and mean reduction of life expectancy = 77%. With these estimated parameters, the model captured the non-linearity in the age distribution and the 10–20 years age group for the peak disease prevalence. The observed OAE prevalence in Maridi was 3.7%, which also showed a strong agreement with model predicted prevalence of 4.07%. Fig 2B shows the prevalence of new OAE onset in the respective age groups. Beyond the 10–20 years of age group, there is no onset of OAE, as observed in the data. For comparative analysis, we also plot the model predictions with the set of parameter values of second and third best fits (see Fig B in S1 Text).

With estimated values of fitted parameters, we plotted OAE prevalence against total mf prevalence. Fig 3 represents a comparison between our model output (prevalence of OAE

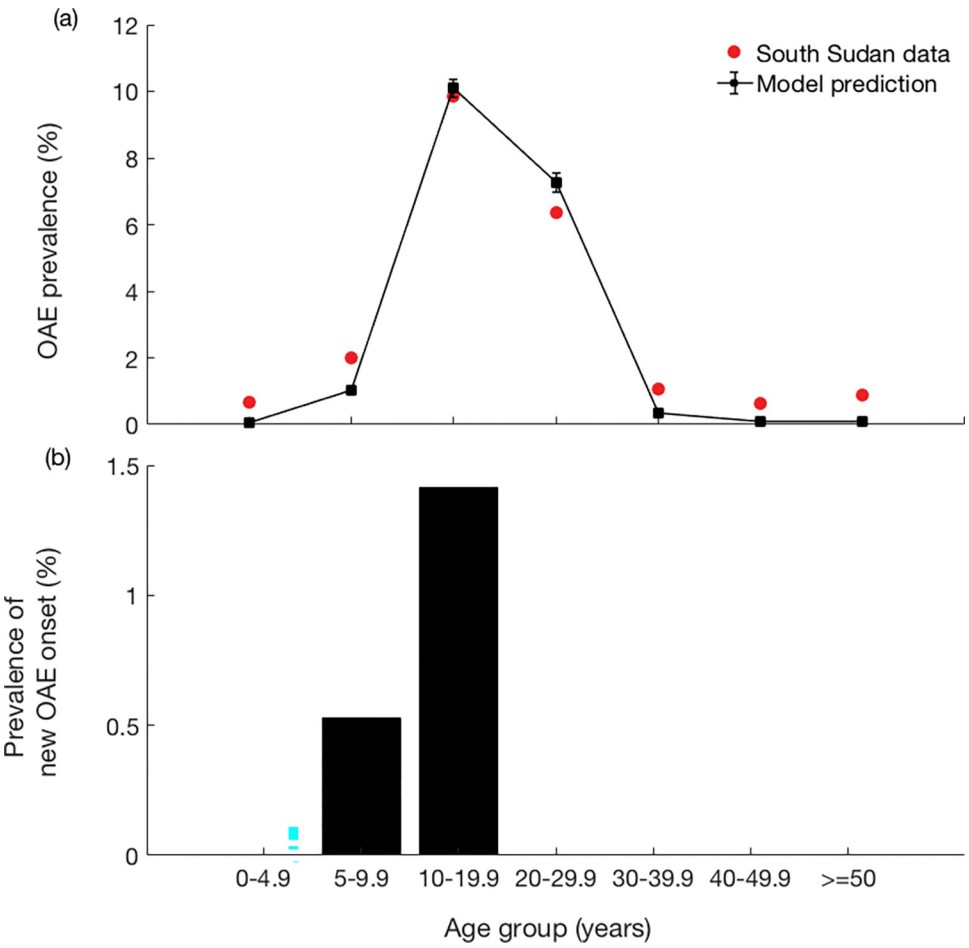

**Fig 2.** (a) Best model fit to the OAE prevalence as observed in 44 villages in 8 study sites in Maridi County in South Sudan in 2018, and (b) new OAE onset obtained from the model in parameter estimation. The black dots represent the mean of 100 runs.

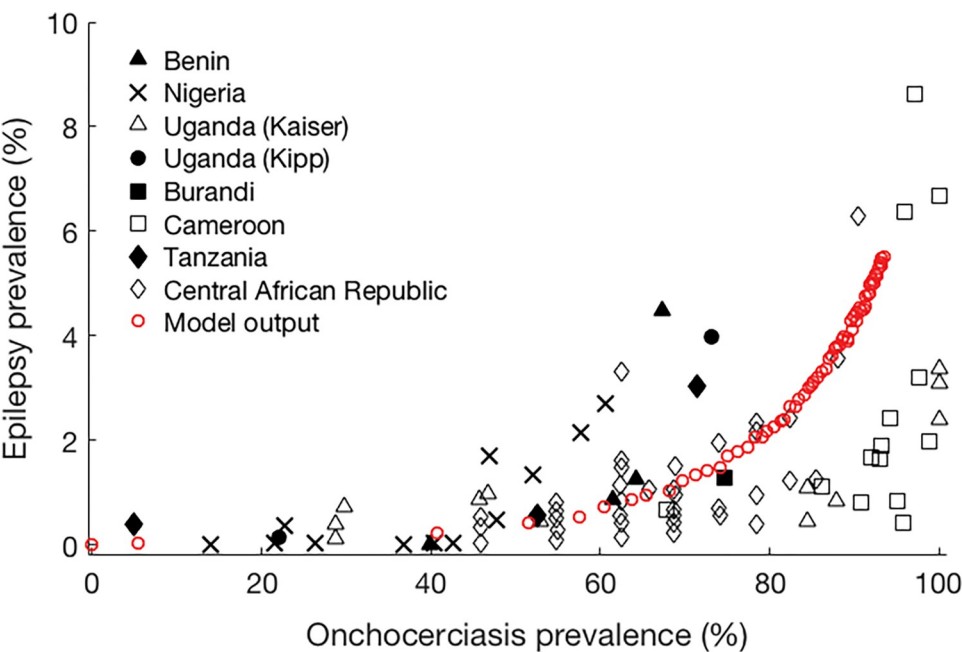

**Fig 3. Model predicted OAE prevalence as a function of total mf prevalence compared with the original survey data obtained from Pion et al. [16].**

only) and results predicted by Pion et al (corrected prevalence of all epilepsies; see Fig 3 in [16]). It shows a very low OAE prevalence (0–2%) when onchocerciasis prevalence remains under 60%, and thereafter the graph exhibits a steeper association. At lower onchocerciasis prevalence levels, the model predicted OAE prevalence is below the overall, which is expected as the model does not capture epilepsy from other causes. At higher onchocerciasis prevalence, the model predicts a steeper rise in OAE prevalence than shown in the survey data (especially from Cameroon), although the predicted level remains well within the range of observed values.

## Sensitivity analysis

We assessed the impact of parameters on fitting model to the data through sensitivity analysis (Fig C in S1 Text). We considered the age of maximum mf accumulation $a_{max}$ at 18 years in estimation of parameters. However, the prevalence peak shifts to the right if we consider the $a_{max}$ lies in between 15–20 years (Fig C(a) in S1 Text). Changing the susceptibility parameter or ABR did not change the basic pattern of OAE prevalence produced by the model (Fig C (b & d) in S1 Text), but higher values of life expectancy reduction decreased the OAE prevalence in older age groups (Fig C(c) in S1 Text).

## Model predicted impact of ivermectin MDA

We used the OAE model with estimated parameters to assess the impact of annual MDA coverage on the OAE incidence and prevalence. We considered three different coverage levels: 50%, 60%, and 70%, and we calculated the incidence from the model output. Fig 4A shows the reduction in OAE cases after multiple years of annual MDA with different population coverages. The OAE incidence also decreased with long-term annual MDA (i.e., number of zeros increased in the curve with higher coverage) (Fig 4B). The OAE prevalence reduced significantly with relatively higher coverage and longer duration of MDA (Fig 4A); with 70% annual

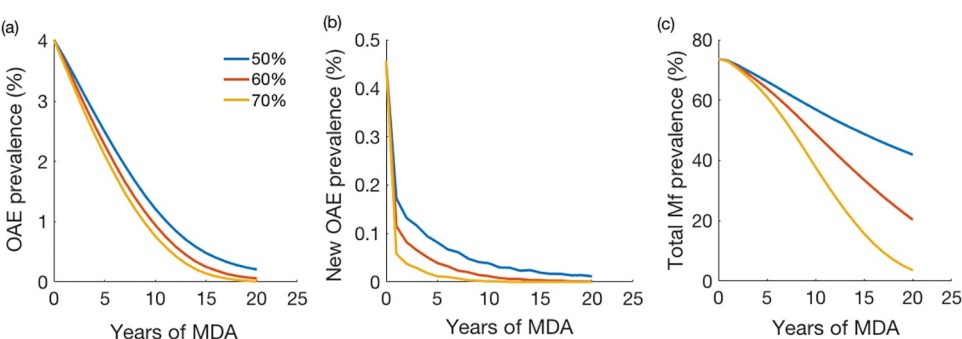

**Fig 4.** Predicted OAE prevalence (%) following the initiation of ivermectin MDA: (a) Incidence of OAE (cases per 100,000), (b) new OAE cases, and (c) total mf prevalence (percentage of population with positive skin snip) under three different coverage levels of annual MDA. Each line represents the mean of 100 simulation runs. Plot of changes in incidence is given in the S1 Text: (See Fig F in S1 Text).

coverage, the prevalence of OAE reduced to almost nil (<0.01%) after 15 years of MDA implementation. The results of multivariate sensitivity analysis of MDA on the OAE incidence are given in Fig D in S1 Text.

Fig 5 shows that there is a clear shift in the mean age of persons with OAE resulting from a change in the age-specific OAE prevalence due to annual ivermectin MDA. Although this pattern is not clearly observed at 50% coverage, with higher coverage of 60% or more, the prevalence in the 11–20 years age group becomes lower as the OAE prevalence peak shifts to the 21–30 years age group (see Fig 2).

## Model predicted impact of vector control

We also explored the effect of vector control with different assumed efficacy (i.e., the assumed percentage reduction of biting rates). OAE incidence declined significantly after the implementation of vector control (Fig 6). In 20 years of vector control with 40% reduction of black-fly biting rates, OAE incidence decreased by 37%, while 80% efficacy achieved approximately 81% reduction of the OAE incidence for the same duration. The results of multivariate sensitivity analysis of vector control on the OAE incidence are given in Fig E in S1 Text.

To predict the impact of combined interventions including annual MDA and vector control, we considered scenarios with three different vector control efficacies: 60%, 70%, and 80% (Fig 7). Three different MDA coverage levels annually were also considered in each scenario: 40%, 50% and 60%. As observed, implementation of annual MDA combined with different efficacy level of vector control for approximately 20 years reduced the OAE incidence significantly to nearly zero.

## Discussion

Using OAE data from Maridi County in South Sudan, we quantified the parameters of a new OAE sub-model in ONCHOSIM. With the estimated parameters, the model generates an age-specific prevalence pattern similar to the empirical OAE data, and consistent with trends observed between onchocerciasis prevalence and epilepsy (all types) as previously reported by Pion et al [16]. We also performed a sensitivity analysis to describe the uncertainty of these parameters when applied to different age groups in our study population. We evaluated the impact of ivermectin MDA under different programmatic scenarios and showed that annual MDA at a good coverage (≥70%) can significantly reduce OAE incidence, so that disease

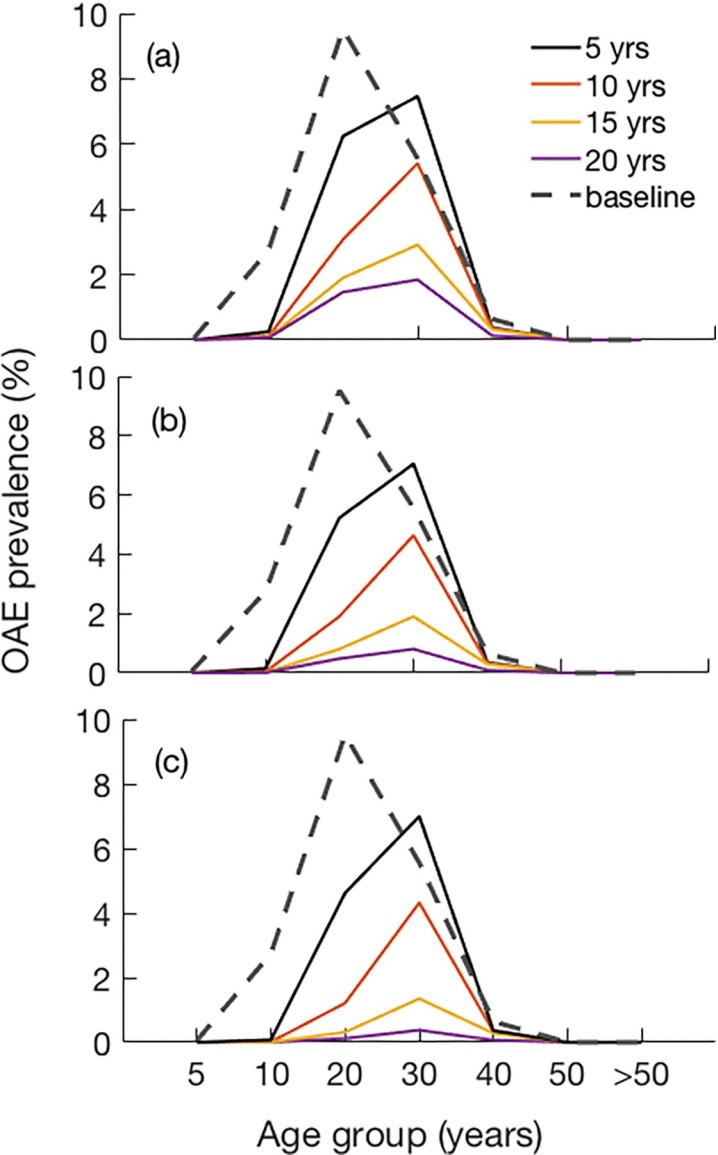

**Fig 5.** Estimated age specific OAE prevalence due to the impact of annual ivermectin MDA with coverages of: (a) 50%, (b) 60%, and (c) 70%.

prevalence shifts towards older age groups. Reducing biting rates via vector control strategies may also reduce the OAE incidence.

Optimal MDA will rapidly affect the OAE incidence by preventing more children and adolescents from developing OAE. Due to the chronic nature of the condition, the prevalence of OAE changes more gradually: people who developed OAE prior to onchocerciasis elimination will remain alive for some time, thereby contributing to the pool of prevalent cases until they die. Our model suggests that MDA to control onchocerciasis control should be implemented consistently for about 10 years in order to notice a substantial impact on the prevalence of OAE. These findings concur with pooled data from West-African countries where epilepsy prevalence was seen to decrease after several years of onchocerciasis control [22]. The decline in prevalence is associated with an age-shift of OAE cases to older age groups as was observed

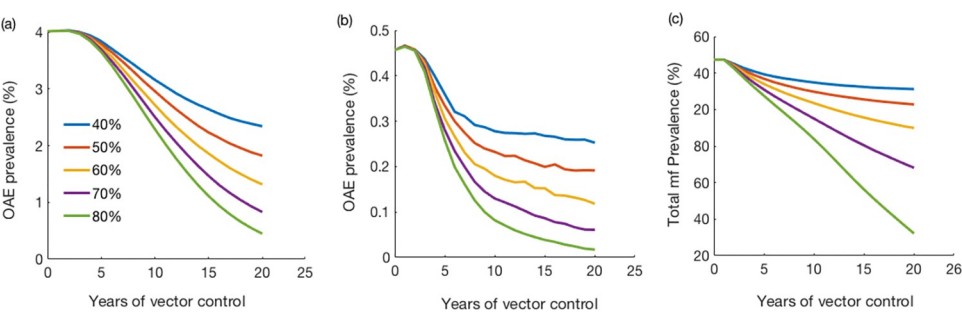

**Fig 6.** Predicted (a) OAE prevalence (%) and (b) new prevalence (%), and (c) total mf prevalence (percentage of population with positive skin snip) after implementation of vector control under five different efficacy scenarios (% reduction in biting rate due to vector control). Plot of changes in incidence is given in the S1 Text (See Fig G in S1 Text).

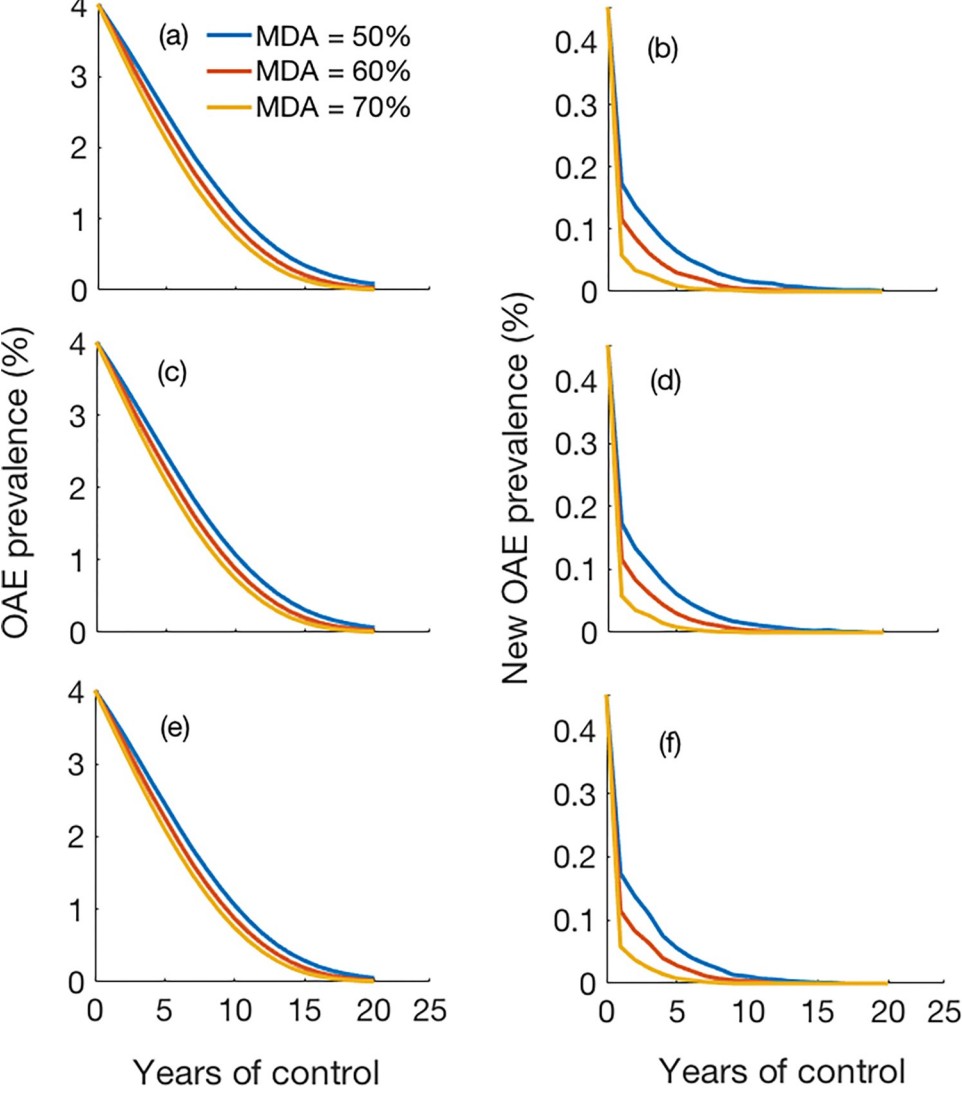

**Fig 7.** Predicted number of OAE cases (Left panel: Total cases; Right panel: new cases) after implementing combined annual ivermectin MDA and vector control of efficacy levels: (a-b) 60%, (c-d) 70% and (e-f) 80%.

in Cameroon and Northern Uganda [21,47]. In our model, ivermectin MDA can reduce the prevalence and incidence of epilepsy, which should motivate policymakers to strengthen onchocerciasis elimination programs, particularly in highly endemic onchocerciasis areas [39].

If there is indeed a causal relationship between onchocerciasis and epilepsy then the current disease burden of onchocerciasis has likely been underestimated so far, as suggested by others [24]. Formerly, the focus of onchocerciasis control programs was primarily on skin and ocular complications of the disease, leading to the premature conclusion that in many onchocerciasis-endemic areas, the disease is no longer a public health problem thanks to long-term interventions [48]. Unfortunately, the neurological manifestations of onchocerciasis such as OAE (including nodding syndrome) cause significant morbidity, disability, and mortality [49]. Preliminary calculations estimated that in former APOC countries, OAE was responsible for an additional 13% of the total reported years lived with disability attributable to onchocerciasis [24]. This additional burden highlights the public health concern that OAE poses to onchocerciasis-endemic communities across Africa. In view of the new targets set by the World Health Organization regarding neglected tropical diseases in general and onchocerciasis in particular [50], highly endemic onchocerciasis areas should be prioritized for the implementation and/or strengthening of onchocerciasis elimination strategies.

## Study limitations

Like any other model, our OAE model has certain limitations. The pathophysiological mechanisms causing OAE remain unclear at present [8]. Hence, it was not possible to explicitly account for more detailed biological processes involved in OAE development and build a mechanistic model of OAE which might provide a better estimation. We have estimated model parameters based on cross-sectional data. A detailed model based on new insights about the pathophysiological mechanism of OAE and with cohort-based individual-level age-stratified OAE data might improve the model.

Despite these knowledge gaps, we have made a first effort in capturing the main patterns in the epidemiology of OAE and how it can be altered by onchocerciasis elimination strategies. We calibrated our model estimates using data from a single location, Maridi County in southern South Sudan, with a rather complex history of control. Further validation is required using age-specific cross-sectional data and longitudinal data from various geographical and epidemiological foci (including other onchocerciasis-endemicity levels). In this regard, a prospective monitoring of the epilepsy and onchocerciasis situation in Maridi County is ongoing to obtain empirical data about the extent to which control interventions against onchocerciasis would impact the incidence of OAE [51]. Such prospective data could serve as a basis to evaluate the performance of our newly developed model in predicting longitudinal OAE trends.

## Conclusion

OAE is a major health problem in foci with high ongoing onchocerciasis transmission in Africa. Although much effort has been invested so far to eliminate onchocerciasis across the continent, the effectiveness has varied and areas with high OAE prevalence remain. Further attention for onchocerciasis elimination is needed in these areas, not only to fight onchocerciasis but also to prevent OAE. Our model results can be used to further optimize onchocerciasis and OAE elimination strategies in areas where current strategies are insufficient. In highly endemic villages in Maridi, South Sudan, we would recommend high ivermectin MDA coverage, particularly among school aged children, together with a low cost and environmental friendly community-based vector control method such as a "slash and clear" strategy [52].

## Supporting information

**S1 Text.** (Section A: Annotated input file; Section B: Plots of estimated sum-squared-values in the parameter space and corresponding prevalence obtained from model; Section C: Sensitivity Analysis and plots; Section D: Multivariate sensitivity analysis of MDA and vector control.) (DOCX)

**S1 Table. It shows how the work adheres to the five principles of the NTD Modelling Consortium.** (DOCX)

## Acknowledgments

We are grateful to the AMREF South Sudan research team for conducting the surveys that provided the data used in this paper.

## Author Contributions

**Conceptualization:** Samit Bhattacharyya, Robert Colebunders.

**Data curation:** Samit Bhattacharyya, Joseph N. Siewe Fodjo, Makoy Y. Logora, Robert Colebunders.

**Formal analysis:** Samit Bhattacharyya, Natalie V. S. Vinkeles Melchers, Joseph N. Siewe Fodjo, Wilma A. Stolk.

**Investigation:** Samit Bhattacharyya, Joseph N. Siewe Fodjo, Robert Colebunders.

**Methodology:** Samit Bhattacharyya, Natalie V. S. Vinkeles Melchers, Wilma A. Stolk.

**Resources:** Amit Vutha, Luc E. Coffeng.

**Software:** Amit Vutha, Luc E. Coffeng.

**Supervision:** Robert Colebunders.

**Validation:** Samit Bhattacharyya, Natalie V. S. Vinkeles Melchers, Joseph N. Siewe Fodjo, Wilma A. Stolk.

**Visualization:** Samit Bhattacharyya, Wilma A. Stolk.

**Writing – original draft:** Samit Bhattacharyya.

**Writing – review & editing:** Natalie V. S. Vinkeles Melchers, Joseph N. Siewe Fodjo, Amit Vutha, Luc E. Coffeng, Makoy Y. Logora, Robert Colebunders, Wilma A. Stolk.

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
