## [Decision Letter · Decision Letter 0]

20 Mar 2023

Dear Dr Bhattacharyya,

Thank you very much for submitting your manuscript "Onchocerciasis-associated epilepsy in Maridi, South Sudan: Modelling and exploring the impact of control measures against river blindness" for consideration at PLOS Neglected Tropical Diseases. As with all papers reviewed by the journal, your manuscript was reviewed by members of the editorial board and by several independent reviewers. The reviewers appreciated the attention to an important topic. Based on the reviews, we are likely to accept this manuscript for publication, providing that you modify the manuscript according to the review recommendations. 

All reviewers highlighted the importance of this study and had only minor requests. Please address their comments.

Sincerely,

Marc P Hubner

Academic Editor

Eva Clark

Section Editor

All reviewers highlighted the importance of this study and had only minor requests. Please address their comments.

Reviewer's Responses to Questions

**Key Review Criteria Required for Acceptance?**

**Methods**

-Are the objectives of the study clearly articulated with a clear testable hypothesis stated?

-Is the study design appropriate to address the stated objectives?

-Is the population clearly described and appropriate for the hypothesis being tested?

-Is the sample size sufficient to ensure adequate power to address the hypothesis being tested?

-Were correct statistical analysis used to support conclusions?

-Are there concerns about ethical or regulatory requirements being met?

Reviewer #1: See summary and general comments

Reviewer #2: This manuscript presents the results of a study that utilizes a well established model for onchocerciasis to investigate the effect of the currently available interventions (ivermectin MDA and vector control) on the prevalence of onchocerciasis associated epilepsy (OAE). OAE has been a manifestation of Onchocerca volvulus infection that has received increasing attention over the past few years. The model used here (ONCHOSIM) is one of the two most well established models for onchocerciasis, and this study involved only minor modifications to the model to evaluate the effect of the interventions on OAE.

Reviewer #3: -The study objectives are clearly articulated,

-the study design is appropriate to address the stated objectives. 

-study population is clearly described,

- the sample size is sufficient to ensure adequate power to address the hypothesis being tested.

-The statistical analysis used support the conclusions.

-There is no concerns about ethical or regulatory requirements

**Results**

-Does the analysis presented match the analysis plan?

-Are the results clearly and completely presented?

-Are the figures (Tables, Images) of sufficient quality for clarity?

Reviewer #1: See summary and general comments

Reviewer #2: The results are clearly presented and are basically what someone knowledgable in the field would have predicted a priori. Ivermectin MDA was predicted to reduce the prevalence of OAE prevalence fairly quickly, while vector control also reduced prevalence, although less rapidly. Combining both MDA and vector control led to the most rapid reduction of prevalence.

I had a few minor concerns that I would like to see the authors address:

1. The authors report that they used data from a single community in South Sudan to parameterize the model and then report that the model closely replicated the observed prevalence of OAE in that community. Using the same dataset to parameterize the model and then validating the model against the same data is circular. The authors only had the single dataset, and recognize this as a limitation in the conclusions. But I do not feel the validation was really sound and suggest deleting this.

2. The authors report the effect of MDA at different coverages, but do not define coverage rates. In the field, two versions of coverage data are used - the percentage of the total population (the APOC standard) and the percentage of the eligible population treated (the Carter Center/OEPA standard). They need to define which of these they used.

3. The authors report the model predictions for OAE prevalence and mention that there is a shift in the prevalence age distribution as a result of OAE being a chronic condition, and reducing the incidence through MDA and vector control would explain this shift. I would actually like to see the model predictions for incidence as well as prevalence.

Reviewer #3: - results presented match the analysis plan

-The results are clearly presented, on page 15, Figure 6: the Y-axis should read New OAE prevalence (%) instead of OAE Prevalence (%)

-Tables and figures are of sufficient quality and clarity

**Conclusions**

-Are the conclusions supported by the data presented?

-Are the limitations of analysis clearly described?

-Do the authors discuss how these data can be helpful to advance our understanding of the topic under study?

-Is public health relevance addressed?

Reviewer #1: See summary and general comments

Reviewer #2: The conclusions are all well supported by the data and the limitations are discussed. Public health relevance is clearly discussed.

Reviewer #3: -The conclusions are supported by the data presented

-the authors have presented the limitations of the analyses performed

- the helpfulness of the findings to advance the understanding of the topic under study is discussed

-the public health relevance of the study is clearly highlighted

**Editorial and Data Presentation Modifications?**

Reviewer #1: See summary and general comments

Reviewer #2: None

Reviewer #3: NA

**Summary and General Comments**

Reviewer #1: Bhattacharyya et al. applied a mathematical modelling (ONCHOSIM) to predict the impact of MDA and vector control on epidemiology of OAE in the Maridi region (South Sudan). Based on their results, the authors suggest that OAE incidence and prevalence can be reduced by onchocerciasis elimination programmes and conclude that these programmes need to be intensified to prevent OAE. The content of the manuscript and presented findings are very important and from broad interest for researchers and health officials/national NTD programmes. However, several questions remain uncertain and need to be addressed:

1) In regards to the model assumptions the authors stated We assume that brain damage is (directly or indirectly) induced by immune responses triggered by dying mf..." Ivermectin is a microfilaricidal drug inducing mf cell death. According to assumption of the authors this would mean that MDA induces brain damage and thus treated individuals have a higher Di (t)/OAE rate?

2) Did the authors consider to include season (rain/dry season) into the model.

3) Since the authors conclude that MDA programmes need to be intensified, would ivermectin treatment twice a year improve OAE incidence.

4) The authors need to check the references. Maybe, I missed it but refs 16 and 24 is not mentioned within the manuscript.

Reviewer #2: This manuscript represents the first attempt to examine the effect of the available interventions for onchocerciasis on OAE. Basically, the results suggest that the current interventions, if successfully implemented, will result in a dramatic reduction in the prevalence of OAE. This is reassuring to those of us involved in the effort to eliminate onchocerciasis in Africa.

Reviewer #3: The development of a mathematical model to predict the impact of the control measures against onchocerciasis is very timing and highly relevant. During the recent years, evidence have accumulated in Sub Saharan Africa on the association between Onchocerciasis and Epilepsy. Studies have also demonstrated that the control of Onchocerciasis has direct impact in reducing the number of new cases of epilepsy in the endemic area of Onchocerciasis. The capacity of predicting the impact of control measures in different scenarios of treatment coverage or Simulium biting rate reduction was lacking to give to the health system a comprehensive tool to address the problem of epilepsy associated to Onchocerciasis. The findings from this study are filling the gap of knowledge.

PLOS authors have the option to publish the peer review history of their article (what does this mean?). If published, this will include your full peer review and any attached files.

Reviewer #1: No

Reviewer #2: Yes: Thomas R. Unnasch

Reviewer #3: Yes: Samuel Wanji

Figure Files:

Data Requirements:

Reproducibility:

References

---

## [Editor Report · Decision Letter 1]

19 Apr 2023

Dear Dr Bhattacharyya,

We are pleased to inform you that your manuscript 'Onchocerciasis-associated epilepsy in Maridi, South Sudan: Modelling and exploring the impact of control measures against river blindness' has been provisionally accepted for publication in PLOS Neglected Tropical Diseases.

Best regards,

Marc P Hubner

Academic Editor

Eva Clark

Section Editor

Thanks for the revised manuscript. You addressed all previous comments from the reviewers and there are no additional requests that require another round of review. However, please make sure to mention the new Supplementary Figure S6 in your final manuscript and the "supplementary information" section.

---

## [Editor Report · Acceptance letter]

23 May 2023

Dear Dr Bhattacharyya,

We are delighted to inform you that your manuscript, "Onchocerciasis-associated epilepsy in Maridi, South Sudan: Modelling and exploring the impact of control measures against river blindness," has been formally accepted for publication in PLOS Neglected Tropical Diseases.

Best regards,

Shaden Kamhawi

co-Editor-in-Chief

Paul Brindley

co-Editor-in-Chief
